



# High resolution Air Quality simulation in the Himalayan valleys, a case study in Bhutan

Bertrand Bessagnet[1], Narayan Thapa[2], Dikra Prasad Bajgai[2], Ravi Sahu[2], Arshini Saikia[2], Arineh Cholakian[1], Laurent Menut[1], Guillaume Siour[3], Tenzin Wangchuk[4], Monica Crippa[5], and Kamala Gurung[2]

[1]Laboratoire de Météorologie Dynamique (LMD), École Polytechnique, IPSL Research University, Ecole Normale Supérieure, Université Paris-Saclay, Sorbonne Universités, UPMC Université Paris 06, CNRS, Route de Saclay, 91128 Palaiseau, France
[2]International Centre for Integrated Mountain Development (ICIMOD), Kathmandu, Nepal
[3]Univ Paris Est Creteil and Université Paris Cité, CNRS, LISA, F-94010 Créteil, France
[4]Jigme Singye Wangchuck School of Law, Paro, Bhutan
[5]European Commission, Joint Research Centre, Ispra, Italy

**Correspondence:** Bertrand Bessagnet (bertrand.bessagnet@lmd.ipsl.fr)

**Abstract.** Our study focuses on Bhutan, a highly mountainous country where governmental authorities are increasingly monitoring air pollution. To support further analysis and the monitoring strategy, we present the first high-resolution air quality simulations with the chemistry transport model WRF-CHIMERE over the western region of Bhutan at a spatial resolution of roughly 1 km. Increasing the horizontal resolution of the model improve the performances, decreases potential errors due to

too important spatial average of meteorological and emissions data having an high spatial variability. However, the air pollutant emissions must be improved at fine scale with better proxy, particularly for industries where improvement are still required. For the first time, we propose high resolution maps of air pollution (concentrations and deposition fields). Our simulations confirm that Bhutan valleys also suffer from air pollution mainly due to $PM_{2.5}$ (concentrations exceeding 20 $\mu g\ m^{-3}$ ) dominated by carbonaceous species, largely above the World Health Organization guidelines. Wildfires and anthropogenic activities release

large amount of carbonaceous species and can also impact the glaciers by atmospheric fallout. Wildfires can locally contribute to 20% of the total $PM_{2.5}$ concentrations over a 15 days period, and theoretically, black carbon can be transported up to the highest peaks. Ecosystems are at risks with deposition fluxes of sulfur and nitrogen species comparable with other locations at risk in the world.

## 1 Introduction

The Hindu Kush Himalaya (HKH) spans over a region particularly affected by air pollution including India, Pakistan, Bangladesh and Nepal which are currently the most impacted by air pollution (HEI, 2025; Mehra et al., 2019). Particularly outside the monsoon season, the combination of favorable meteorological conditions and large emission sources in the Indo-Gangetic Plain is the main reason of impressive outbreak of pollution events. In the region, air pollution not only affects health (HEI, 2025). Air pollution also has an effect on ecosystems (loss of biodiversity) through the deposition of inorganic species like ammonia



(Beachley et al., 2024) as explained by Bhagowati and Ahamad (2019), and on the melting of glaciers (Gul et al., 2021; Kang et al., 2020) due to deposition of Black Carbon on snowy surfaces. Particles have also an effect on meteorological conditions and implications on the development of the monsoon in the region (Santra et al., 2025). Environmental risks in HKH related to climate change directly or indirectly affect around 2 billion people in Asia.

Bhutan, the smallest country in the region, is bordered to the north by China and to the west, south, and east by India. The
country is crisscrossed by numerous rivers that flow southward, creating a number of deep valleys where the most significant cities are located. These include the capital, Thimphu, which is at 2,400 meters above sea level, Paro, which has the country's only international airport, and Haa, which is close by and reachable from Paro via a pass at 3,900 meters above sea level. The elevation varies from more than 7,000 meters in the north to 200 meters in the south near the Indian border. Bhutan is considered as a pristine environment and looks less affected by air pollution compared to its neighbors. Bhutan is covered by
70% of forest, and a particularity of this country is the exposure of its population to indoor air pollution mainly due to cooking activities and heating systems using wood (Wangchuk et al., 2015; Pratali et al., 2019; Khumsaeng and Kanabkaew, 2021).

Traditional wood-burning stoves, called *Bukhari* are extensively used for heating and cooking (Wangchuk et al., 2017). These residential combustion sources in deep valley has an effect on ambient air concentrations, they represent more than 80% of primary $PM_{2.5}$ (Particulate Matter with diameter below 2.5 $\mu m$) emissions according to the EDGAR (Emission Database for
Global Atmospheric Research) emission database (Crippa et al., 2024; Guizzardi et al., 2025). Along the year, $PM_{2.5}$ concentrations range on a monthly basis from less than 20 $\mu g\,m^{-3}$ during the monsoon to more than 40 $\mu g\,m^{-3}$ in wintertime (Sharma et al., 2021), largely above the WHO (World Health Organization) guidelines of 5 $\mu g\,m^{-3}$ for $PM_{2.5}$ (WHO, 2023, 2021). The Royal Government of Bhutan is a leading country to fight against air pollution, the Thimphu Outcome summarizes the key discussions and recommendations from the Second Regional Science Policy Dialogue on Air Quality Management in the Indo-
Gangetic Plain and Himalayan Foothills (IGP-HF) held on June 26-27, 2024, co-organized with ICIMOD and the World Bank (ICIMOD and World Bank, 2024).

Western Bhutan's air quality is often deteriorated due to continuous forest fires during pre-monsoon, and then climate change may place rural livelihoods at risk (Vilà-Vilardell et al., 2020) with more and more favorable conditions leading to outbreak of fires. In the region the effect of forest fires on air quality has been studied by Kumari et al. (2024) indicating the urgent
need for targeted interventions to mitigate the impact of forest fires on air quality in the North-East of India close to Bhutan. According Karthik et al. (2022), in India, vegetation burning contributes more than 80% of carbon stock. In Bhutan a recent study (Sharma et al., 2022) estimated the potential source regions contributing to the $PM_{2.5}$ concentrations in Thimphu during the years 2018–2020 using a Lagrangian model. They showed that 80% of $PM_{2.5}$ were due to external sources mainly coming from India. However, for the country, there is a need to set-up a more robust and comprehensive modeling platform to analyse
the role of each sources to tailor the more appropriate mitigation strategies and enhance a regional dialogue to collectively curb air pollution in the region.

The use of models in such an area is challenging and combines several difficulties related both to the reconstruction of air pollutant emission inventories at high resolution and the simulation of meteorology in such a mountainous zone (Singh et al., 2024). The issue of the spatial resolution of models to simulate the BC deposition fluxes was highlighted by Kang et al.





(2020) who considered crucial to increase their spatial resolution. Indeed, so far, models are mainly applied at coarse resolution around 10 km with emission inventories at similar resolutions. A recent study over Europe shows the added value of using high-resolution simulations to assess the impact on health and potential social inequalities, as emission reduction strategies have highly spatially heterogeneous consequences (Pisoni et al., 2025). This type of studies needs to be extended over the whole South-Asia, and furthermore we must prepare high resolution simulations in the region to train super-resolution models

based on deep-learning techniques to address the air quality at the urban scale over wide domains with complex topography as initiated by Bessagnet et al. (2021).

While Ciarelli et al. (2025) recently focused at 1 km resolution over Nepal to estimate the role of nucleation in the formation of ultrafine biogenic particles, our study is the first to evaluate the air quality and deposition fluxes of key atmospheric species

with a chemistry transport model with a 1km resolution in several Himalayan valleys of the West of Bhutan, Haa, Paro, Thimphu and Punakha. The objective of our study is fourfold: (i) evaluate the performance of the CHIMERE model against available ground observations and set-up a stable framework, (ii) propose a first cartography of the potential impact of air pollution, (iii) evaluate the impact of forest fires and (iv) along the analysis, evaluate the role of the spatial resolution of the model in such a complex topography area.

## 2  Model set-up

The chemistry transport model CHIMERE (Menut et al., 2024) coupled with the meteorological model WRF (Weather Research Forecast) meteorological model (Skamarock et al., 2008) is used to simulate the air pollutant concentrations over the 10 February to 31 March 2025 period. Five days prior to this period are used as a spin-up to initialize the concentration fields in the model. The period corresponds to a measurement campaign in Haa to evaluate indoor and outdoor exposure. This post-winter

period is still cold in Bhutan with some morning frosts, and temperatures which can exceed $15^oC$ in the afternoon. As other locations in the Himalaya, the diurnal cycle of the winds is governed by strong daytime up-valley winds and weak nighttime winds (Potter et al., 2018; Mikkola et al., 2023). Katabatic winds in these areas are expected to have consequences on the transport of air pollutants from the highest layers of the atmosphere (Yang et al., 2015). Wildfires are commonly observed from March in Bhutan that is the beginning of the pre-monsoon period. Some light precipitations (rain and snow) in the valleys are

observed during the studied period in the Western Bhutan.

The CHIMERE model is a regional chemistry-transport model that can be used in both online and offline configurations in its latest version for research, scenario analysis and operational forecast purposes (Lapere et al., 2021; Bessagnet et al., 2020; Menut et al., 2020). Three domains are designed targeting the West part of Bhutan at $0.01^o$ resolution (Figure 1 and 2). The model needs a set of gridded data as mandatory input: emission data for both biogenic and anthropogenic sources, land use

parameters, boundary and initial conditions, and other optional inputs such as dust and fire emissions. Given these inputs, the model calculates the concentrations and wet/dry deposition fluxes for a list of gaseous and aerosol species (depending on the chosen chemical mechanism).

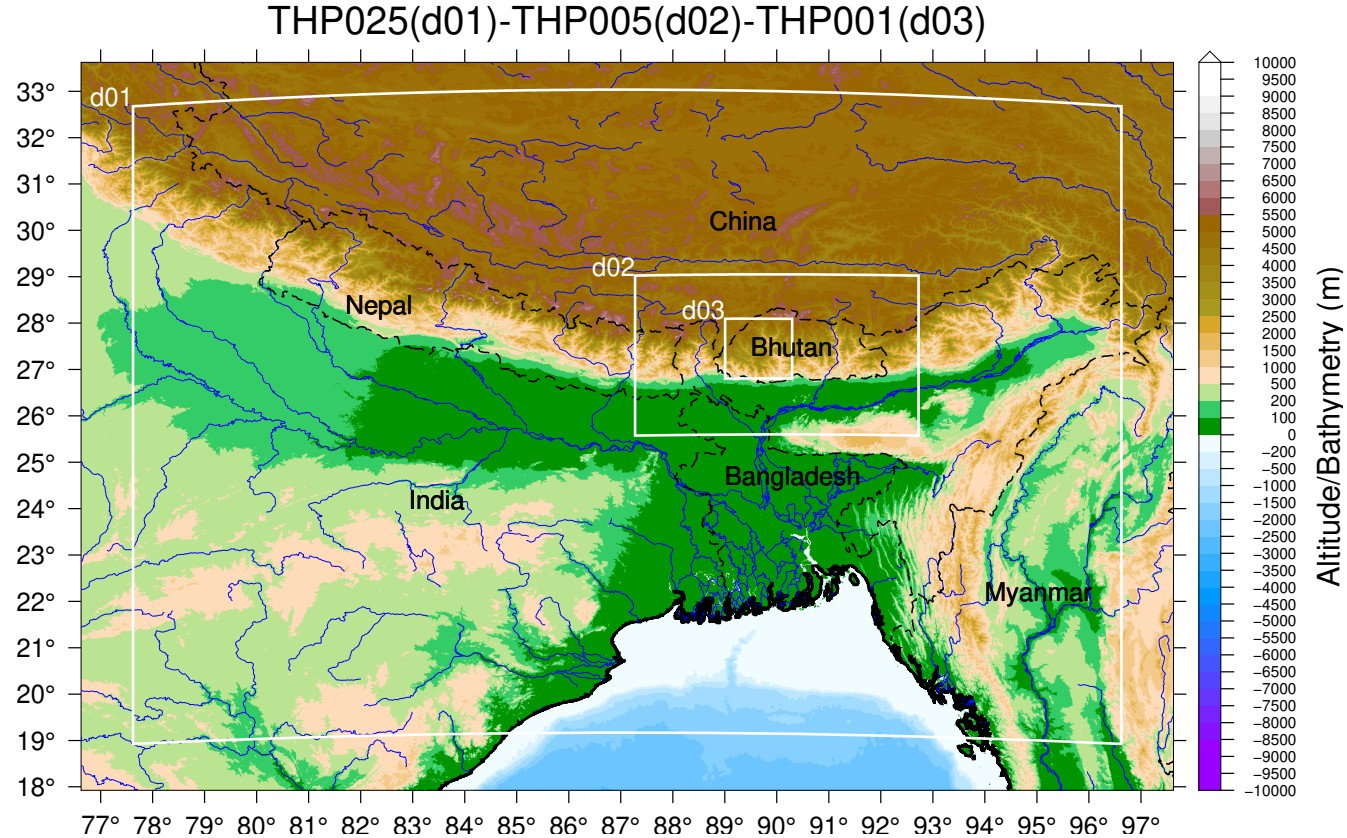

**Figure 1.** Simulation domains (white frames) in our study

In this study, the model is coupled with WRF using the NCEP raw data (National Centers for Environmental Prediction) at $1^o x 1^o$ (Kalnay et al., 1996) for the global meteorological conditions and initialization. The WRF-CHIMERE suite is run on a triple nested configuration, with a coarse domain covering the HKH at a $0.25^o$ x $0.25^o$ resolution (THP025/d03), the intermediate domain over the whole Bhutan with a $0.05^o$ x $0.05^o$ resolution (THP005/d02), while the finest domain focused on the West Bhutan region with at a $0.01^o$ x $0.01^o$ resolution (THP001/d03). As described in section 3, we have developed a fine emission inventory at $0.01^o$ x $0.01^o$ resolution so that the three simulations longitude/latitude regular domains are a multiple of the emission grid: $\times 1$ for THP001, $\times 5$ for THP005 and $\times 25$ for THP025 (Figure S1). The CHIMERE vertical resolution contains 20 layers starting from the surface going up to 200hPa. For the WRF configuration we have increased the resolution with 46 eta-levels until 30hPa to account for the complex topography. Fire emissions were used from the CAMS Global Fire Assimilation System (CAMS, 2022). Spectral nudging is applied for the coarse domain THP025 only nested domains (also within the Planetary Boundary Layer). The wind components, the potential temperature perturbation and the water vapor mixing ratio are nudged with a relaxation coefficient $g = 0.0003\ s^{-1}$. A wave number of 5 and 4 is used, respectively on $x$



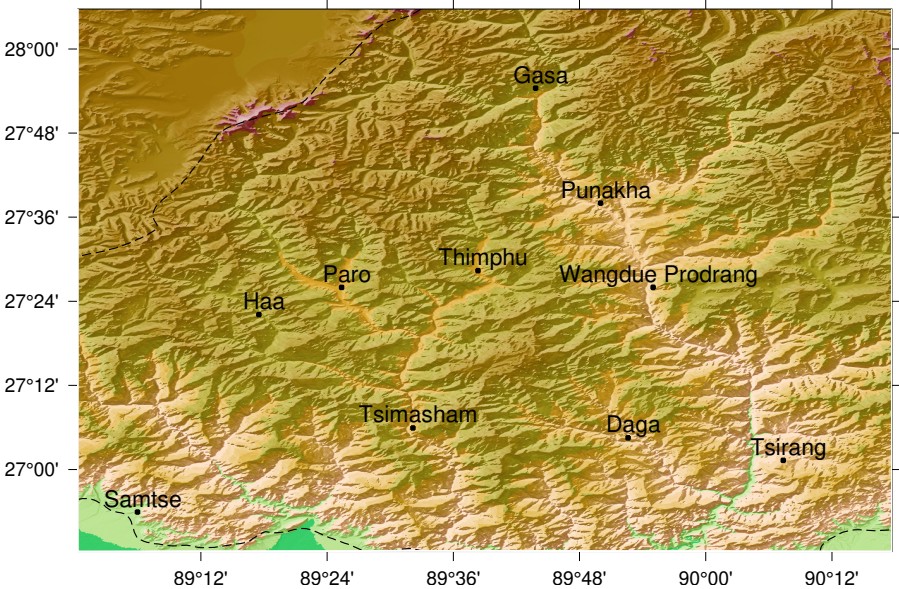

**Figure 2.** High resolution domain THP001/d03 over Western Bhutan. The highest mountains (Jomolhari –7326m– and Jichu Drake –around 6800m–) are located over the North-West part of the domain along the border with China. The main valley are cleary identified, Haa being a high altitude valley.

and $y$ directions. For this exercise, all simulations are made off-line without feedback (radiative effects) of aerosols on the meteorology.

The resuspension scheme is activated within the domain (Vautard et al., 2005), it does not include road trafic resuspension. The resuspension process is important for particulate matter and may induce a large increase of the emission flux in case of dry soils, for locations where traffic and industries produce available particles. This resuspension process is considered as different

and complementary of the aeolian erosion process which is not activated for our domains (no relevant arid zones in the region of interest). Biogenic VOC emissions are computed online with the MEGAN 2.10 algorithm (Guenther et al., 2012). Boundary conditions were taken as monthly climatologies from LMDz-INCA (Hauglustaine et al., 2014) as standard data provided in the CHIMERE package. For mineral dust, we use the monthly climatologies at the boundaries from the GOCART model which is more adapted for our analysis on dust deposition since we want to capture an average behaviour for a month (section 6.2).

Dust episodes are very sporadic with a strong temporal variability, as explained in Vautard et al. (2005), the monthly median value of dust as boundary conditions is used instead of the average concentration.

All major aerosol groups are activated in the model including elemental carbon, organic matter, sulfate, nitrate, ammonium, SOA (Secondary Organic Aerosols), mineral dust (from the boundaries), sea salts, and PPM (here non-carbonaceous Primary Particle Matter and resuspended particles); taking into account coagulation, nucleation and condensation processes over 10

size bins ranging between $10nm$ - $40\mu m$. The chemical formation of secondary organic aerosols (SOA) from primary organic aerosol is not activated here. Wet scavenging and dry deposition is considered following the Wesely's parameterization (Wesely,





1989). For more details, the WRF and CHIMERE configuration files are provided in Table S1 and S2 in the supplementary material.

## 3 Anthropogenic emissions

There is no available high resolution air pollutant emissions inventory in the region. We use the procedure developed by Bessagnet et al. (2023) to create an emission inventory at fine scale ($0.01^o$ resolution) from a coarse resolution emission dataset. This methodology builds a high resolution inventory based on proxy for each activity sector of a coarse emission database. Our coarse emission inventory at $0.1^o$x$0.1^o$ is the EDGAR database (Crippa et al., 2024) for the year 2022 and the main pollutants CO, $PM_{10}$, $PM_{2.5}$, Organic Matter assuming $OM = 1.6 \times OC$ according Philip et al. (2014), and accessible

at https://edgar.jrc.ec.europa.eu/dataset_ap81, Black Carbon (BC), $SO_2$, NOx, $NH_3$, NMVOC (Non methane Volatile Organic Compounds). Primary OM and BC is considered in the fine fraction $PM_{2.5}$. We have then developed a $0.01^o \times 0.01^o$ over the whole South-Asia for the same pollutants. For the residential sector (RCO) and the traffic emissions (TRO), the downscaling has been applied from the country level to the fine grid. For the industrial and all other remaining sectors we prefer to use the downscaling from the EDGAR grid to benefit from a first spatialisation embedding more information at sub-national level.

Moreover, EDGAR data are provided over 36 possible classes of activity sectors for annual emissions and 8 macro sectors for monthly emissions (Table S3 of supplementary material). Therefore, we have calculated monthly profiles over the 8 macro sectors and applied them to the 36 possible sub-categories to maximize the use of information.

Bhutan's emissions reflect its economy based on agriculture, livestock, forestry, mining, the sale of hydroelectric power to

India, and tourism. An extraction of Bhutan emissions from the EDGAR database (Table 1) shows the major contribution of residential emissions (RCO) for most of compounds. NOx is mainly emitted by the residential, industrial and traffic sectors. One peculiarity of Bhutan comes from this residential sector which is the most emitter either for NOx and $NH_3$ whereas in other neighbor countries like India, traffic and agriculture respectively contribute the most for these two pollutants. Agriculture is often cited as the main source of ammonia but wildfire emissions is probably underestimated as mentioned by Felix et al.

140 (2023).

The global population database GHSL (Global Human Settlement Layer) developed by the European Commission - Joint Research Centre (Pesaresi et al., 2024) is used. These data at 100 m resolution were used to downscale the emissions from the residential sector (mainly heating, air conditioning and cooking operation).

Specific emissions from butter lamps and incense burning (Yangzom et al., 2024) in the context of religious rituals in Bhutan

are not taken into account here. Another source related to road dust resuspension is particularly important in wintertime and pre-monsoon period. For the residential sector while for gas emissions we have directly used the population density to spatially reallocate the emissions, we use a different approach for particulate emissions. Indeed, these latter emissions are mainly emitted by wood burning mostly occurring in rural places. The methodology calibrated over Europe (Terrenoire et al., 2015) with bottom-up emissions gave a population based proxy $px^p$ for particulate species as a function of population density




**Table 1.** Emission of air pollutants for Bhutan according EDGAR (kton in 2022) for main activity sectors reconstructed from the gridded data

| EDGAR sectors[*] | $PM_{10}$ | $PM_{2.5}$ | BC | OC | CO | NMVOC[†] | NOx | $NH_3$ | $SO_2$ |
|---|---|---|---|---|---|---|---|---|---|
| AGS | 0.034 | 0.022 | - | - | - | 0.161 | 0.211 | 2.202 | - |
| AWB | 0.336 | 0.320 | 0.029 | 0.213 | 3.667 | 0.350 | 0.147 | 0.128 | 0.019 |
| CHE | 0.001 | 0.001 | - | - | - | 0.010 | - | 0.003 | 0.002 |
| ENE | 0.049 | 0.045 | 0.004 | 0.003 | 0.876 | 0.077 | 1.351 | 0.011 | 2.328 |
| FOO_PAP | 0.015 | - | - | - | - | 0.074 | - | - | 0.023 |
| IND | 1.091 | 0.977 | 0.218 | 0.171 | 6.645 | 1.682 | 1.390 | 0.263 | 2.573 |
| IRO | 0.101 | 0.101 | - | - | 0.202 | 0.005 | 0.015 | - | 0.007 |
| MNM | 0.088 | 0.015 | - | - | - | 0.296 | 0.021 | 0.408 | - |
| NMM | 0.406 | 0.295 | 0.009 | - | 0.001 | - | 0.001 | - | 0.324 |
| PRO_FFF | - | - | - | - | - | 0.287 | - | - | - |
| PRU_SOL | 0.004 | 0.004 | - | - | - | 2.113 | - | - | - |
| RCO | 34.19 | 17.47 | 2.16 | 7.873 | 222.3 | 34.84 | 2.678 | 6.443 | 1.763 |
| REF_TRF | 2.246 | 0.068 | 0.027 | 0.003 | 4.602 | 2.982 | 0.002 | 0.174 | - |
| TNR_Aviation_CDS | 0.002 | 0.002 | - | - | 0.011 | 0.004 | 0.107 | 0.001 | 0.009 |
| TNR_Aviation_CRS | 0.003 | 0.003 | 0.001 | - | 0.020 | 0.007 | 0.202 | 0.003 | 0.016 |
| TNR_Aviation_LTO | 0.001 | 0.001 | - | - | 0.030 | 0.001 | 0.097 | 0.001 | 0.009 |
| TNR_Other | 0.021 | 0.021 | 0.003 | 0.002 | 0.193 | 0.025 | 0.223 | - | 0.054 |
| TNR_Ship | 0.097 | 0.097 | 0.019 | 0.010 | 0.911 | 0.163 | 0.441 | - | 0.120 |
| TRO | 0.070 | 0.055 | 0.020 | 0.016 | 2.305 | 0.788 | 1.503 | 0.021 | 0.002 |
| WWT | - | - | - | - | - | 0.001 | - | 0.004 | - |
| *Total* | *38.75* | *19.49* | *2.492* | *8.293* | *241.7* | *43.92* | *8.390* | *9.662* | *7.248* |

[*] *AGS: Agricultural soils, AWB: Agricultural waste burning, CHE: Production of chemicals, ENE:Energy, FOO:Food Production, IND:Combustion in manufacturing industry, IRO: Iron and Steel production, MNM:Manure Management, NMM: Production of non metallic minerals, PRO_FFF: Fuel Exploitation, PRU_SOL: Solvents, RCO: Small scale combustion, REF:Refineries, TNR_Aviation _CDS: Aviation climbing & descent, TNR_Aviation_CRS:Aviation cruise, TNR_Aviation_LTO: Aviation landing & takeoff, TNR_Aviation_SPS: Aviation supersonic, TNR_Other: Railways, pipelines, off-road transport, TNR_Ship: Shipping, TRO:Road Transport, WWT: Waste Water Treatment.*

[†] *Non Methane Volatile Organic Compounds*

*pop* where the coefficient c is set to 1.5 here after several trials (Equation 1).

$$px^p = pop \times [\log_{10}(pop)]^{-c} \tag{1}$$

In short, the proxy is used to calculated the high resolution emission $e$ in a high resolution grid cell $i$ as in Equation 2 for a downscaling of emission $E$ of the coarse level structure $C$ which is either the country or the corresponding coarse grid cell.

$$e_{i \in C} = E_C \times \frac{px_i}{\sum\limits_{j \in C} px_j} \tag{2}$$

This formula indicates that PM emission *per capita* is more important in rural places due to more frequent use of wood for heating and cooking (Denier van der Gon et al., 2015). This behavior is very usual worldwide, households use the resource that is the less expensive and very accessible in their close environment. The use of this methodology allows to differentiate wood




versus non-wood gas and particle emissions. As reported in the European Environment Agency guidebook on air pollutant emissions (EEA, 2023), the PM/NOx emission factor ratio is the largest for wood used in all stoves compared to LPG (Liquefied

Petroleum Gas) or other fossil fuels burnt under more controlled combustion processes.

For most other proxy, the OSM (OpenStreetMap) project (OpenStreetMap contributors, 2017) is used and rasterized at 3 arc second (about 90m). Harbors, airports, industries, crops can then be spatially reallocated at the adequate resolution. We calculate a 3" arc seconds proxy which represents the percentage of surface occupied by a land cover. These proxy are then aggregated at $0.01^o$ resolution in the downscaling procedure. For roads, to estimate the surface, a specific width in meters is

assigned for each type of roads as: *unclassified*: 9m, *motor*: 15m, *primary*: 15m, *secondary*: 12m, *tertiary*: 7m, *residential*: 5m, *footway*: 3m, *service*: 5m. A pre-treatment is operated at ⅓ arc second (about 10 m) before reaggregation at 3 arc seconds.

As shown in Figure 3, we can clearly see the added value of the downscaling procedure for two major activity sector. In the EDGAR raw database at coarse resolution the main road in Bhutan cannot be identified while at fine resolution the reallocation of the emissions provides a more realistic picture at $0.01^o$ resolution. For instance, at coarse resolution the roads in Haa valley

did not appear.

## 4   Available observations

In Bhutan, there are very few air quality monitoring stations, and only the Thimphu Air Quality Monitoring Station is partially functional. It is currently operated by the Department of Environment and Climate Change, Bhutan, with the support of ICIMOD. The station employs reference-grade (state-of-the-art) equipment's, including Thermo Scientific (USA) instruments

for trace gases ($O_3$, NOx, CO, and $SO_2$), a Grimm (Germany-based) instrument for PM ($PM_{10}$ and $PM_{2.5}$), an Aethalometer (Slovenia) for Black Carbon (BC), and Lufft for meteorological data. In the Haa measurement, we have used two Air Beam Low-cost sensors in two different locations (Katsho and Uesu) for $PM_{2.5}$.

In this study, we have considered the observation data from 20 February 2025, to 10 March 2025, for the Thimphu air quality and from 22 February 2025, to 26 March 2025, for Haa Valley in Bhutan (Figure S2-S4). All stations are assumed to be urban

or peri-urban background stations *i.e.* supposed to be not influenced by local sources. Their location is reported in (Table 2). Some meteorological stations have been considered for the evaluation of model performances.

**Thimphu (Capital city of Bhutan)**

The measurement data show substantial temporal variations in PM and gaseous pollutants. The daily (24-hour) concentration

of $PM_{2.5}$ is consistently above the WHO guideline of 15 $\mu$g m$^{-3}$ in Thimphu, with more than double for most of the day. The BC concentration shows that the sharp episodic peaks reaching 15 $\mu$g m$^{-3}$ and above, with a background concentration below 5 $\mu$g m$^{-3}$ . The correlation between $PM_{2.5}$ and BC is very strong ($R^2$=0.91), it indicates common combustion sources. Hourly $PM_{10}$ baseline background concentration ranges from 20 to 40 $\mu$g m$^{-3}$ , and some peaks exceed 100 $\mu$g m$^{-3}$ . The $PM_{2.5}$ concentration shows a similar pattern, but with generally lower values, ranging from 10 to 50 $\mu$g m$^{-3}$ , with occasional peaks

exceeding 50 $\mu$g m$^{-3}$ .





**Figure 3.** Impact of the downscaling for two species - primary PM$_{2.5}$ and NOx - from low (left) to high (right) resolution for two activity sectors : residential combustion (RCO - top) and traffic (TRO - bottom) over the west Bhutan.

For Ozone, the running 8-hour average maximum concentration remains consistently above the WHO guidelines and the Bhutan National Ambient Air Quality Standard 2020 (51 ppb) throughout the study period, with the exception of one day (28 March). It shows a marked diurnal cycle, with hourly concentrations ranging from 3 ppb at night to 78 ppb in the afternoon. The lowest concentration was observed on 28 February when there was light rainfall, clouds, and no solar radiation. The inverse

relationship between the NOx (NO$_2$, NO), CO, and O$_3$ clearly illustrates that the atmospheric reaction and enough precursor compounds are available in the atmosphere (Bhutan) for ozone formation. The NOx illustrates the distinct diurnal patterns with peaks during the morning (8:00 to 10:00) and evening. The daily variations perfectly align with the high traffic hours patterns. The hourly concentration ranged from 1.5 ppb to about 20 ppb during the measurement periods, and with daily concentration ranged from 3 ppb to 9 ppb. CO concentration displays clear diurnal variability with baseline levels around 200-300 ppb

and episodic peaks exceeding 600 ppb. NOx and CO show a strong temporal correlation ($R^2 = 0.88$) during the measurement periods, which indicates a significant contribution from common traffic emission sources to the ambient concentrations.



**Table 2.** Location of stations with observations (air quality and/or meteorology)

| Name–District (Dzongkha) | Code | Longitude ($^o$E) | latitude ($^o$N) |
|---|---|---|---|
| Thimphu Capital–Thimphu | THP | 89.635 | 27.472 |
| Babesa–Thimphu[†] | BAB | 89.384 | 27.254 |
| Natgsho–Haa[†] | NAT | 89.262 | 27.403 |
| Katsho–Haa[*] | KAT | 89.277 | 27.392 |
| Uesu–Haa[*] | UES | 89.311 | 27.351 |
| Airport–Paro[†] | PAR | 89.422 | 27.405 |

[†] Official meteorological stations

[*] Stations equipped with low cost sensors

$SO_2$ concentrations are generally low during the monitoring period, typically remaining below 0.6 ppb, with occasional minor peaks. It indicates that no major emission sources for $SO_2$ are located in the region.

**Haa (High altitude Valley in Bhutan)**

The use of Low Cost Sensors (LCS) is common in South-Asia (Shabbir et al., 2025) and an hybrid approach with LCS and reference stations is probably the key to improve the air quality monitoring. For Haa, we have used two Air Beam Low-cost sensors developed by HabitatMap in two different locations (Katsho and Uesu) only for $PM_{2.5}$. It is a laser-based light scattering optical particle counter technique to measure the PM concentration, temperature, and relative humidity. The instrument records data at intervals of 1 minute in its internal memory. It transmits data via Wi-Fi to a cloud platform for remote access. Before starting the measurements, these sensors are collocated in the Thimphu station, and during data calibration, a correction factor is applied. Based on the measurement, the 24-hour average $PM_{2.5}$ concentration is consistently higher than the WHO air quality guideline (15 $\mu$g m$^{-3}$ ) throughout the 33-day measurement period. During this period, the daily concentration ranges from 20 to 55  $\mu$g m$^{-3}$ in Katsho and from 17 to 57 $\mu$g m$^{-3}$ in Uesu. The highest concentrations were recorded from 18 to 21 March at the measurement sites. To evaluate the model, we also used meteorological data measured by the LCS and official stations in Bhutan for the surface temperature (T2M), relative humidity (RELH) and the wind speed (WINS).

## 5   Model performances

Global error statistics (RMSE as Root Mean Square Error, Correlation and bias) on daily data are presented in Table 3, the mean diurnal cycles are displayed in Figure S5 as well as all timeseries for all pollutants in Figure S6 and S7. As in many studies using regional models (Bessagnet et al., 2016) the PM concentrations are underestimated, certainly due to emission inventories which underestimate some emissions like wood burning, a major source in Bhutan difficult to estimate. For the $PM_{10}$ concentrations, the model underestimates the afternoon concentrations with an increase from 2 to 4:00 pm local time



while at the same time PM$_{2.5}$ does not increase. This feature could be related to a missing emission of coarse PM emissions like road dust resuspension (emissions due to the flow of vehicles).

Indeed, in South-Asia, outside the monsoon period, the road traffic resuspension is a major source of coarse traffic-related particles. In Delhi, a recent study showed that 79% of these particles came from resuspension of road dust (Singh et al., 2020). The contribution of non-exhaust PM emissions will get more significant since exhaust emissions will decrease with the renewal of the vehicle fleet and these emissions are not taken into account in models. For the Thimphu station a very good correlation coefficient of 0.85 exists between the observed PM coarse fraction as $(PM_{10} - PM_{2.5})/PM_{10}$ by using a quadratic regres-

sion. The absolute time correlation reaches 0.96 for the diurnal cycle (Figure S8). The coarse fraction is correlated (linear regression) with the wind speed with a much weaker correlation coefficient of 0.45 showing that urban resuspension is not the main driver. The relative humidity is therefore a good predictor of the resuspension which is for a urban station likely due to road dust resuspension. Considering the resuspension negligible near 100%, a remaining 10% PM coarse fraction is probably due to other anthropogenic emissions. For very low humidity, the coarse fraction of 60 to 80% is of the same order

of magnitude found by Singh et al. (2020) in Delhi. In addition, Amato et al. (2012) consider that the road dust resuspension also affects the fine fraction, 25% of re-suspended dust can be considered in the fine fraction.

 As displayed in Table 3 and Figure 5, there is clearly an improvement of statistics when we increase the resolution, but from $0.05^o$ to $0.01^o$ we obtain mixed results. Indeed, as mentioned in the literature (Schaap et al., 2015; Colette et al., 2014;

Terrenoire et al., 2015), for the concentration of pollutants, increasing the resolution can also produce numerical noise and can degrade some statistics. The largest improvement is obtained from $0.25^o$ to $0.05^o$ however the bias is often at the highest resolution. Valari and Menut (2008) showed for ozone an optimal resolution at about 10 km, which overcomes the problem of uncertainty in wind characteristics at too high resolution when comparing on sparse observational datasets, however this study was performed over the Paris region with a smooth topography. Bessagnet et al. (2020) showed a similar result in a mountainous

region with the main improvement from 10 to 3 km spatial resolution for most pollutants with still an improvement from 3 to 1 km on the bias only.

 For the gases, only observations at the station of Thimphu are available. Ozone concentrations are overestimated, particularly during nighttime (Figure S5) and certainly a consequence of underestimation of NO$_2$. It is usual for regional models to overestimate ozone at coarse resolution because of mesh-related numerical effects (Gao et al., 2025). The temporal correlation

of Ozone is low in Thimphu station because of lack of variability in February-March and the model overestimate the concentrations particularly at night. The Thimphu station is located not too far from a busy road and some burning activities have been reported explaining the negative bias of the model on BC as well. Indeed, from 19 February a jump of BC concentrations is observed due to locals sources while the model was more in line with observations from 10 to 18 February with even a slight positive bias. The diurnal cycle of BC shows the two peaks in the morning and afternoon is respectively shifted in the model

by 1 and 2 hours.

 As already observed in studies using models driven by global emission inventories, it is usual to overestimate sulfur dioxide (Pachon et al., 2024). We remind here that sulfur dioxide concentrations are driven by the industrial and energy sectors, and




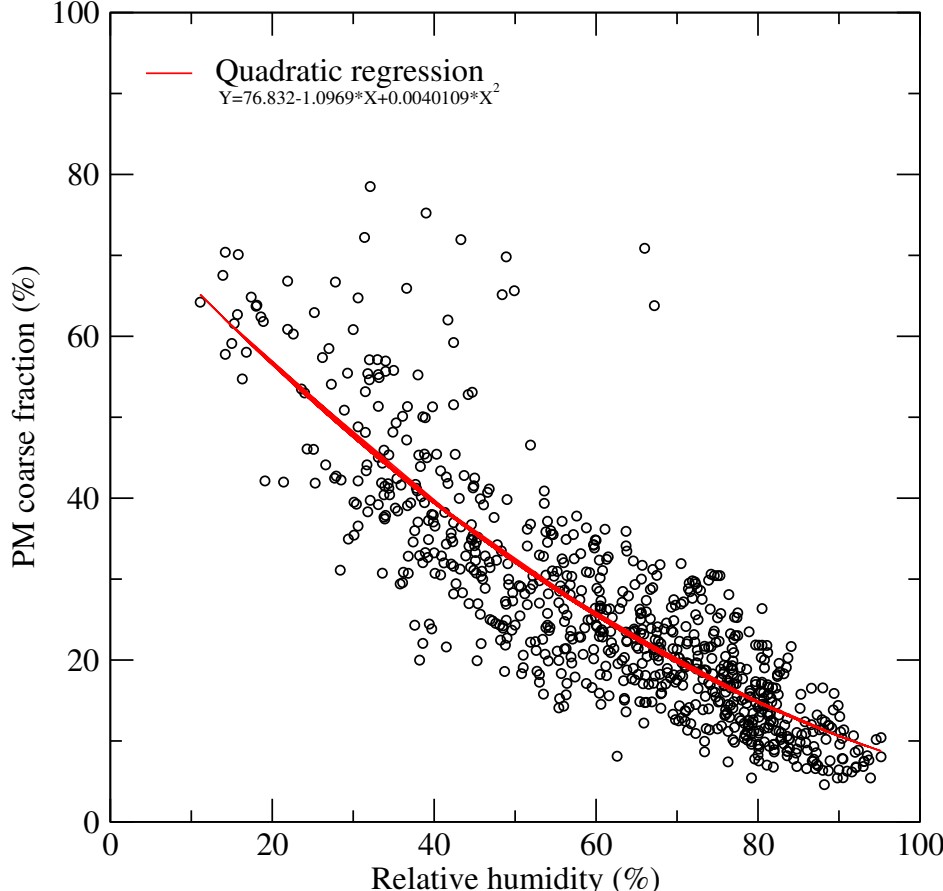

**Figure 4.** Coarse PM fraction at Thimphu air quality station as a function of relative humidity (circles) and the corresponding quadratic regression (in red)

moreover the high resolution inventory is downscaled from the gridded EDGAR database. These facilities are typically located outside major urban areas and the proxy used by EDGAR and the downscaling approach could be not adapted leading to these discrepancies. This has been demonstrated by (Thunis et al., 2022) by using a screening approach analysing the inconsistencies that arose on spatial distribution differences for the industrial and energy sectors.

The simulation of meteorological variables is improved at the highest resolution and particularly for the nighttime temperatures (Figure S5). Indeed, within the valleys the prediction of the inversion layers and cold pools require an increase of the model spatial resolution (Bessagnet et al., 2020). The wind speed is underestimated during the afternoon, however the diurnal cycle is well captured by the highest resolution simulation. For the relative humidity, the high resolution helps to retrieve the right time of the lowest values in the afternoon.





**Table 3.** Spatio-temporal error statistics based on hourly values from 10 February to 31 March 2025 at three resolutions: $0.25^o$ (THP025), $0.05^o$ (THP005) and $0.01^o$ (THP001). The RMSE and bias are expressed in the following units: $\mu g\ m^{-3}$ for PM and BC concentrations, ppb for gas concentrations, and °C, %, m s$^{-1}$ respectively for the 2 meter temperature (T2M), relative humidity (RELH) and wind speed (WINS). The last two columns are respectively the averaged observed values (Obs.) and the number of observations (#)

| | *Correlation (%)* | | | *RMSE* | | | *Bias* | | | *Obs.* | *#* |
|---|---|---|---|---|---|---|---|---|---|---|---|
| | $0.01^o$ | $0.05^o$ | $0.25^o$ | $0.01^o$ | $0.05^o$ | $0.25^o$ | $0.01^o$ | $0.05^o$ | $0.25^o$ | | |
| PM$_{2.5}$ | 40 | 39 | 21 | 17.31 | 16.17 | 18.86 | -9.82 | -8.29 | -12.21 | 30.2 | 2199 |
| PM$_{10}$ | 24 | 30 | 3 | 28.29 | 23.24 | 29.92 | -6.42 | -2.09 | -21.67 | 42.2 | 649 |
| BC | 9 | 2 | 19 | 2.80 | 2.67 | 3.12 | -1.41 | -1.27 | -2.36 | 3.2 | 650 |
| NO$_2$ | 15 | 26 | 36 | 4.02 | 3.46 | 4.42 | -2.51 | -2.18 | -3.70 | 4.9 | 433 |
| O$_3$ | 31 | 36 | -7 | 22.04 | 21.65 | 24.66 | 15.32 | 15.15 | 16.70 | 44.0 | 433 |
| SO$_2$ | -10 | 18 | -13 | 1.74 | 1.93 | 0.23 | 1.15 | 1.48 | 0.09 | 0.2 | 433 |
| T2M | 40 | 40 | 41 | 6.21 | 9.17 | 10.71 | -2.70 | -7.09 | -9.26 | 10.5 | 4054 |
| RELH | 54 | 51 | 17 | 18.55 | 22.39 | 28.19 | -8.56 | 7.68 | 13.69 | 62.3 | 4054 |
| WINS | 59 | 55 | 26 | 3.98 | 4.11 | 4.56 | -1.32 | -1.22 | -1.21 | 4.0 | 2504 |

# 6 Air pollution and its impact in maps

## 6.1 Concentrations fields

As mentioned in the introduction, Black Carbon is a key component of PM for Health and climate concerns, in this section

we analyse the spatial patterns of BC and and PM$_{2.5}$ (Figure 6). Over the Indo Gangetic Plain we found values from 2 to 5 $\mu g\ m^{-3}$ on average for BC that is in line with values reported in the literature (Romshoo et al., 2023; Smaran and Vinoj, 2024). The highest values are observed in India and Bangladesh as well as in other big cities like Kathmandu (Nepal). The influence of Delhi (outside the domain) can be observed on the west part of the domain through the monthly climatology used as boundary conditions here.

Focusing over Bhutan, the PM air pollution is concentrated within the valleys with a maximum in Thimphu, An other maximum is observed in the south which corresponds to the location of two important cities: Pasakha in Bhutan and Jaigaon in India. This place is a major industrial zone and this hot spot of air pollution is mainly driven by the industrial sectors. Close to the highest PM$_{2.5}$ values, the contribution of Black Carbon can exceed 5%. BC concentrations are generally below 1 $\mu g\ m^{-3}$ in mountainous sites in line with observations reported in many high altitude stations (Singh et al., 2023). Over the

Tibetan Plateau, average BC concentrations are low between 0.1 and 0.3 $\mu g\ m^{-3}$ as reported in recent studies (Wang et al., 2024).

In the Himalayan region, the fine fraction is mainly made of Secondary Inorganic species (SIA) and primary carbonaceous species (CARB), this latter species being very dominant in the IGP region including the south of Bhutan (Figure 7). Over the



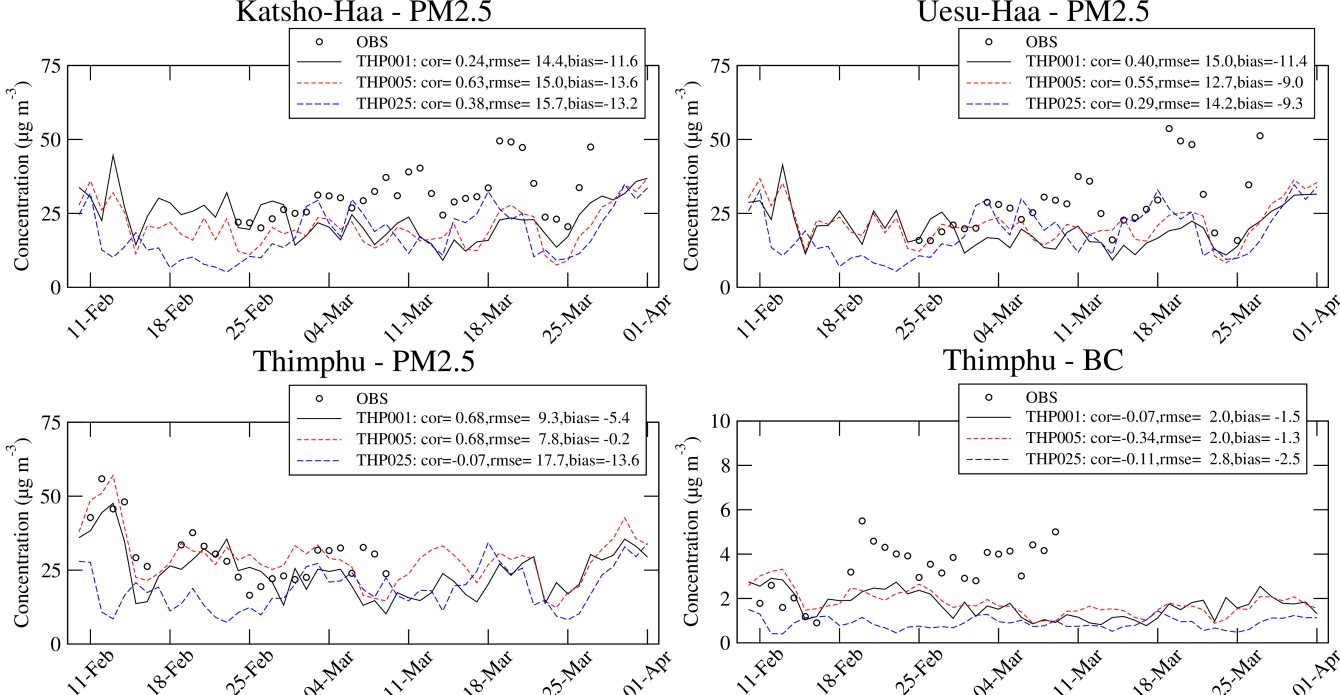

**Figure 5.** Model *versus* observations comparisons and error statistics at stations for the three resolutions for PM$_{2.5}$ and BC, based on daily values

north, mineral dust is also the main component where the total PM$_{2.5}$ is low, below 5 $\mu$g m$^{-3}$ . In Figure S9 is displayed
the same type of figure for the ultra fine fraction of PM (below 0.1 $\mu$m) where SIA largely dominates everywhere, whereas
non carbonaceous primary PM dominates (OPPM) in the south of the domain for the coarse PM (diameter between 2.5 and
10 $\mu$m). There is no studies looking at the composition at the regional level of PM$_{0.1}$ in South-Asia. In Europe, Argyropoulou
et al. (2025) finds high concentrations of SIA (with high sulfate concentrations due to nucleation) and organics in the ultrafine
fraction of PM. In our split of species, organics are split between SOA and CARB categories with a likely underestimation
of secondary production since in our chemical mechanism the production of SOA from the emitted organic aerosol was not
activated. This assumption is supported by recent studies in India (Panda et al., 2025; Bhattu et al., 2024) showing the chemical
production of secondary organics from biomass burning and traffic emissions.

### 6.2 Deposition spatial patterns

The cumulative total deposition for the following list of macro species are displayed in Figure 8 for the month of March 2025:

– *N(red)*: total reduced nitrogen species: ammonia (NH$_3$) and ammonium (NH$_4^+$),

  – *N(oxi)*: total oxidized nitrogen species: NOy as NO+NO$_2$+HNO$_3$ NO$_3^-$ and other PAN species (Peroxyacetyl nitrate),



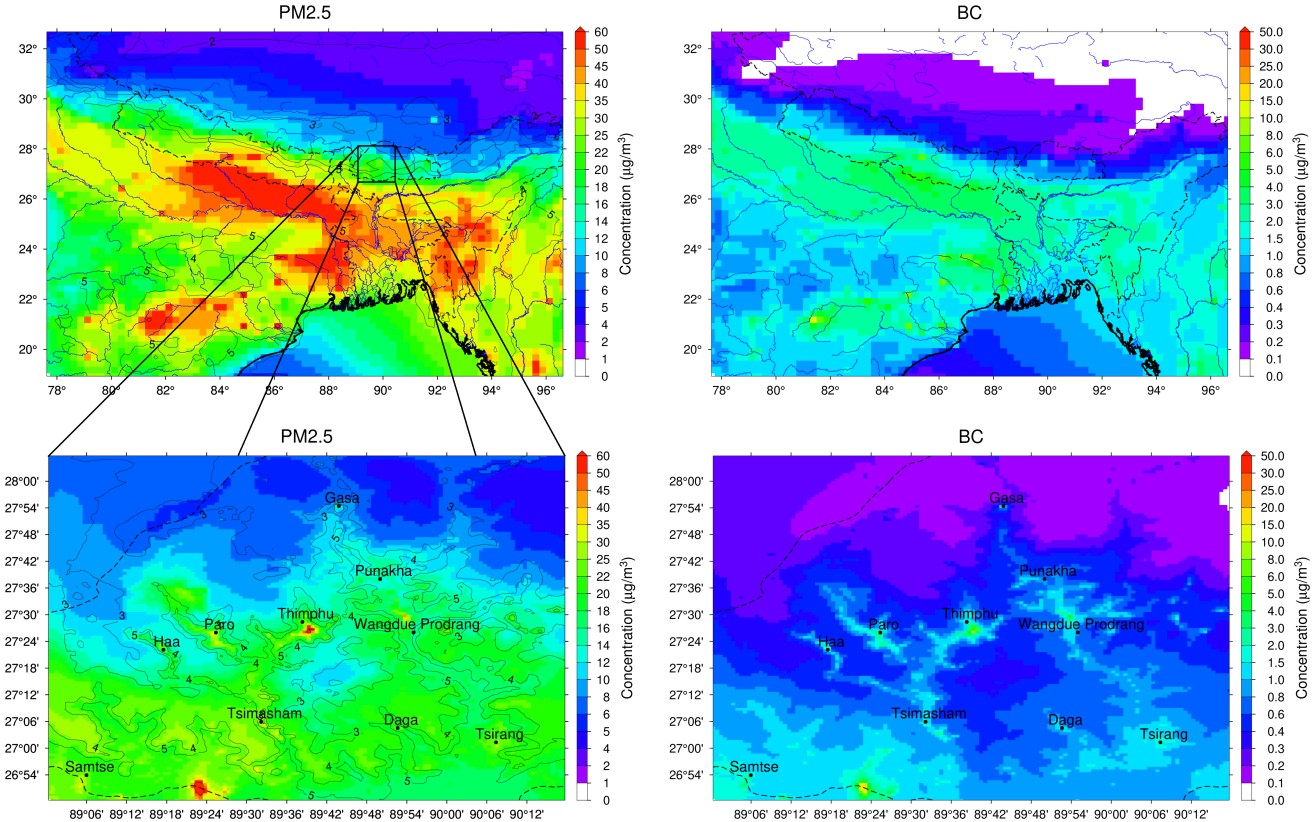

**Figure 6.** Average spatial patterns of PM$_{2.5}$ and BC concentrations in March 2025 over the Hindu Kush Himalayan region (top) and the West Bhutan (bottom). For the PM$_{2.5}$, contour lines represent the percentage of BC in the PM$_{2.5}$ (color scales are different for PM$_{2.5}$ and BC).

- *S(oxi)*: total oxidized sulfur species: sulfur dioxide (SO$_2$) and sulfate (SO$_4^{2-}$),

- *C(bc)*: carbon from black carbon,

- *C(oc)*: carbon from the organic matter issued from primary species (mainly wood burning and wildfires here),

- *Dust*: from mineral dust emissions (here transported from boundaries of the domains)

Carbon from the secondary organic species is excluded in *C(oc)* to mainly target the role of wood burning and wildfires.

**Carbon and mineral dust**

One of the most significant light-absorbing substances causing the atmosphere to warm is black carbon (BC). After its release

into the atmosphere from the burning of fossil fuels and biomass, BC can settle on distant glacier surfaces and hasten glacier melting, leading to increased negative mass balances of glaciers and diminished freshwater supplies downstream (Li et al.,





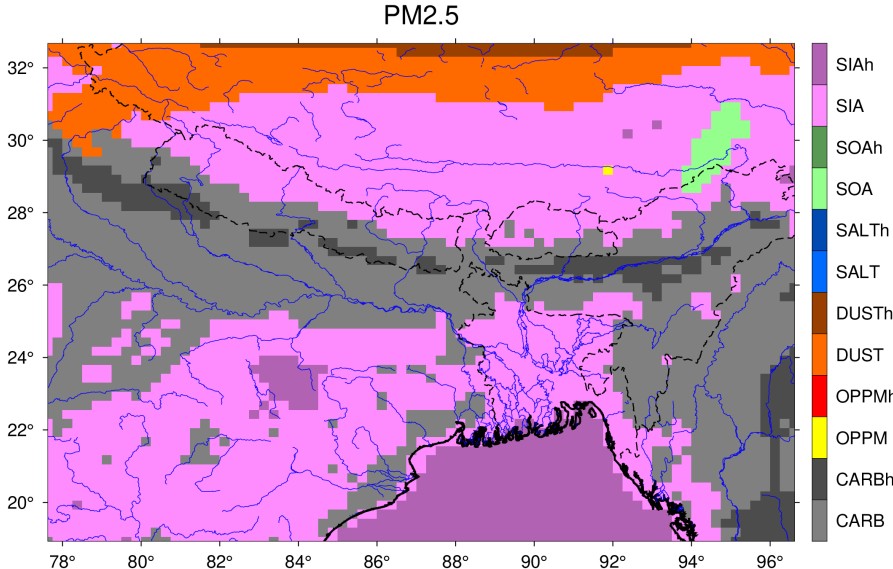

**Figure 7.** Average dominant components in the fine fraction of PM (PM$_{2.5}$) over the widest domain THP025 in March 2025. Macro species are named as CARB (BC and primary OM), OPPM (other mineral primary anthropogenic species), DUST (mineral desert dust), SALT (sum of sodium and chloride), SOA (Secondary Organic Aerosol) and SIA (Secondary Inorganic Aerosol as the sum of nitrate, sulfate and ammonium). Suffix $h$ mentions when the dominant species concentration is at least twice the second most important component.

2023; Réveillet et al., 2022; Lapere et al., 2023; Li et al., 2021). Therefore, when researching the climatic effects of BC in the atmosphere and in the glacier regions, it is crucial to take into account three key links: BC and mineral dust emissions from source regions, transport, and deposition in remote glacier regions. The darkening of snowy surface is probably underestimated by chemistry transport models if the Brown Carbon (BrC), a fraction or Organic Matter mainly emitted by biomass burning, is not well taken into account in models (Chelluboyina et al., 2024). There are also evidence in the literature that mineral dust have a strong impact on glacier melting (Chandel et al., 2025), and even could dominate the darkening of glaciers through the long range transport of dust after sand and dust storms.(Sarangi et al., 2020). Therefore, it is of major importance to also track the primary fraction of the organic matter as a good tracer of the BrC.

In our study, over our domain covering the West Bhutan, we observe that the primary carbon deposition fluxes is mainly due to organic matter –C(oc)– largely emitted by biomass burning (Figure 8 and Table 4). Then, BrC deposition fluxes in Bhutan are probably higher than for BC. Over the highest mountain of the domain close to the border with China (*Jomolhari* peak), total BC deposition range from 100 to 300 $\mu$g m$^{-2}$ in March while C(oc) deposition is 10 times higher from 1 to 2 g m$^{-2}$. Gul et al. (2024) found a monthly rate of BC dry deposition close to 140 $\mu$g m$^{-2}$ using another regional model for another Himalayan glacier in the Himalaya (Yala in Nepal) which is coherent with our results and probably overestimated as stated by the authors. It is also important to better understand what we call BC in emission inventories, whether it is BC (based on



**Figure 8.** Spatial patterns of total deposition in March 2025 over the highest resolution domain THP001 at 0.01$^o$. The contour lines show the wet deposition fraction (25%, 50% and 75%).

optical measurement) or Elemental Carbon as EC (based on thermal methods). At emission BC is probably higher than EC and after ageing in the atmosphere and depending on the mixing state of particles Carbon could have different optical properties





(Liu et al., 2022; Zhang et al., 2023). We also must differentiate which fraction of carbon comes from biomass burning to be
treated as BrC with less absorbing optical properties. In the air pollution community, these clarifications are crucial.

Regarding dust, similarly to the findings of (Sarangi et al., 2020), dust deposition is much larger than black carbon deposition
and will have an impact on the glacier melting (Figure 8 and Table 4). For our simulation, there is no emissions of dust in our
domains and, overall, the fraction of wet deposition over total deposition is below 5% for the period of interest. Therefore, all
deposited dust come from outside the domains though a complex interaction of vertical mixing processes within the boundary
layer involving mountain breeze, sedimentation and then deposition over forested areas and over the glaciers as well. From
the southwest side of the domain, the long range transport of dust in the free troposphere above 4,000 m a.s.l. impact first the
higher hills stretching from Tsimasham to the North-West (see Figure 8). The cross section in Figure 9 displays the transport
of dust from the southwest to the northeast of the finer domain for a given date in March showing a high altitude transport of
dust at 5,000 m early morning which will be mixed later within the development of the planetary boundary layer. From these
findings, it comes that is important to include dust, BC and BrC to study the impact of air pollution on glaciers melting.

Focusing on deposition over the glaciers and perpetual snows (last rows of Table 4), the deposition fluxes for carbonaceous
species have the same order of magnitude as for the whole domain but the fraction of wet deposition is much larger showing
the importance of wet deposition over the glaciers due to precipitations and a less efficiency of dry deposition over bare soils.
For dust, the deposition fluxes are twice lower over the glaciers compare to the average value over the whole domain but it
remains comparable.

**Inorganic species (Sulfur and Nitrogen)**

From large scale models at low resolution, the global picture of ammonia patterns show that South-Asia is a hot spot (Pai
et al., 2021; Xu et al., 2018). Very few data exist over the Himalayan region regarding the deposition of inorganic species
like reduced and oxidized nitrogen species. Nearby, in east Asia, a network monitors the acid deposition and air pollution
(UNEP and ACAP, 2025). Wang et al. (2023) have reconstructed a positive trend of ammonium deposition over the Tibetan
glaciers since 1950 but a slight decrease since 1990 probably due to changes of atmospheric circulations. Using the isotopic
composition of nitrogen, Bhattarai et al. (2019) showed that the air contamination from South Asia to Himalayan Tibetan
Plateau is very likely impacting the high altitude ecosystems. A most recent and exhaustive study has been performed by
(Ellis et al., 2022). They first evaluated published literature defining nitrogen thresholds (critical levels and loads) at which
lichen epiphytes are impacted. Second, to characterize model variability, they employed estimates from previously published
atmospheric chemistry models up to 10km resolution projected to the Himalaya with different spatial resolution and timelines.

We find that nitrogen reduced species deposition dominates with values close to 50 mgN m$^{-2}$ accumulated in March 2025
(Figure 8). For oxidized nitrogen values are slightly lower between 10 and 30 mgN m$^{-2}$. The south of the domain centered in
West Bhutan as well as the valleys are the most affected, however even if agriculture is not the major source of ammonia in
Bhutan, the reduced nitrogen can be transported over very long distance as ammonium in particles and can be then deposited
very far from primary ammonia emission. The average value of the total deposition flux of nitrogen for the month of March
2025 (around 300 gN ha$^{-1}$ as reported in Table 4) is comparable with the values reported in (Ellis et al., 2022) showing





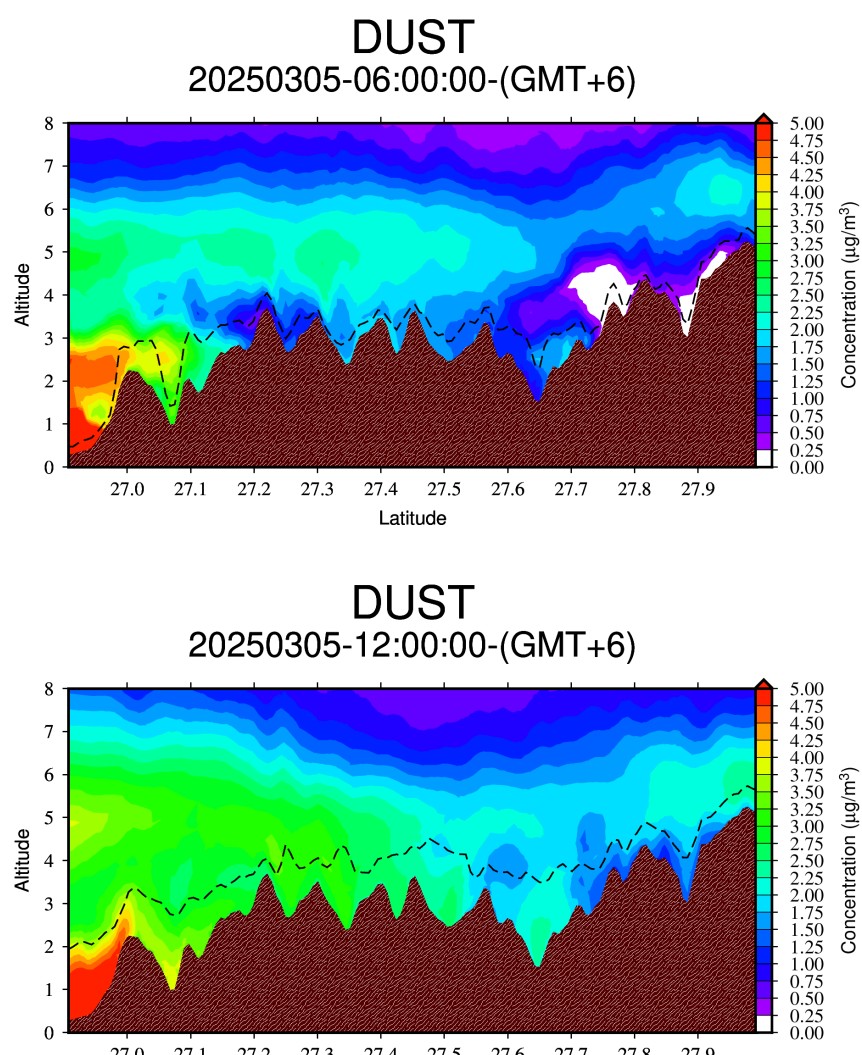

**Figure 9.** Vertical cross section of dust concentrations from the Southwest to the Northeast over the THP001 domain ($89^{o}$E-$26.9^{o}$N to $90.15^{o}$E-$28^{o}$N) the 5 March 2025 at 06:00-GMT+6 (top) and 06:00-GMT+6 (bottom). The dash-line represents the average height of the planetary boundary layer.

observed or modelled values above 2 kgN ha$^{-1}$ yr$^{-1}$ in many locations in the world. For Europe, deposition values of more than 1 kgN ha$^{-1}$ yr$^{-1}$ were simulated by (Vivanco et al., 2017). As elsewhere in the world, deposition of oxidized sulfur is lower than the deposition of total nitrogen compounds. The highest values are located in the south of the domain over the most industrial areas.






**Table 4.** Impact of the horizontal resolution on the average deposition flux simulated by CHIMERE of key species – N(red), N(oxi), S(oxi), C(bc), C(oc) and dust – over the highly resolved domain THP001 at $0.01^o$ resolution. Last rows are average values of deposition fluxes focusing on the glaciers and perpetual snows for C(bc), C(oc) and dust

|  | Deposition flux (g ha$^{-1}$) | | | Wet Deposition (%) | | | Dry Deposition (%) | | |
|---|---|---|---|---|---|---|---|---|---|
|  | $0.25^o$ | $0.05^o$ | $0.01^o$ | $0.25^o$ | $0.05^o$ | $0.01^o$ | $0.25^o$ | $0.05^o$ | $0.01^o$ |
| *Over the whole domain* | | | | | | | | | |
| N(red) | 164.9 | 205.7 | 199.8 | 6 | 5 | 6 | 94 | 95 | 94 |
| N(oxi) | 95.2 | 98.7 | 98.3 | 2 | 2 | 3 | 98 | 98 | 97 |
| S(oxi) | 79.0 | 107.3 | 92.8 | 3 | 3 | 4 | 97 | 97 | 96 |
| C(bc) | 5.5 | 6.0 | 5.9 | 14 | 23 | 29 | 86 | 77 | 71 |
| C(oc) | 24.9 | 26.9 | 26.3 | 14 | 25 | 31 | 86 | 75 | 69 |
| Dust | 135.9 | 172.5 | 173.9 | 3 | 4 | 4 | 97 | 96 | 96 |
| *Over the glaciers only* | | | | | | | | | |
| C(bc) | 4.5 | 3.7 | 3.9 | 56 | 76 | 78 | 44 | 24 | 22 |
| C(oc) | 21.0 | 18.0 | 20.5 | 54 | 76 | 80 | 46 | 24 | 20 |
| Dust | 91.2 | 62.1 | 82.5 | 10 | 20 | 21 | 90 | 80 | 79 |

In Table 4 is highlighted the effect of the spatial resolution ($0.25^o$, $0.05^o$ and $0.01^o$) on the total deposition flux for each species computed over the finest domain THP001 focused on West Bhutan. While the calculation of deposition is properly done

by looping over each land use, we calculate the fluxes over the glaciers as a post-treatment by considering only the location of glaciers and their surface fraction in each cell. Chemistry transport modelling involved various non-linear processes, the main being the chemical production of species like ozone and secondary inorganic PM (like ammonium nitrate). Moreover, the landuse fraction and the meteorology changing with the change of resolution will imply a difference on the calculation of deposition fluxes. Finally, accumulating over one month, the difference on the total amount is not so important considering all

these non-linearities and input data variability.

However, oxidized sulfur and reduced nitrogen species look the most affected by the resolution since they are involved in very non-linear process where the spatial resolution will play an important role. For instance, the sulfur chemistry depends on the pH in clouds which is affected by the the resolution where cloud can be not diagnosed in a coarse grid while they could be formed at a finer grid. In March, dry deposition dominates the total deposition which is normal during this post-winter period

with low recorded precipitation amounts. The wet deposition increase with an increase of resolution, as the consequence of an increase of precipitation with higher resolution simulations (30% more precipitation in THP001 compared to THP005). The wet deposition fraction is significantly higher for carbonaceous species than for inorganic species. It means that these carbonaceous species are probably emitted close to location affected by precipitations. Inorganic species are issued and formed close to more urbanized or industrialized areas father away from remote places.





## 7 Evaluation of the wildfire contribution to air pollution


Biomass burning is known to have an impact up to the Tibetan Plateau (Li et al., 2016). Wildfires are observed in the north of Haa and Paro districts (Dzongkha) during three main periods in March 2025: 6-8, 17-21 and 26-31 (Figure 10). Wildfires were confirmed by VIIRS Fire and Thermal Anomalies available from the NOAA-21 satellite (NASA VIIRS Land Science Team, 2020) as shown in Figure S10. For the first period, the wildfire started at a middle distance between Paro and the Jomolhari

peak and the 3D video supplement (powered by VAPOR – Sgpearse et al., 2023) based on CHIMERE outputs shows a potential impact up to the glaciers surrounding the peak at concentration lower than 1 $\mu$g m$^{-3}$ .

A combination of several processes make the transport of BC possible: (i) the daytime anabatic flows along the valley, (ii) the increase of the planetary boundary layer, and (iii) finally the concentrations are cleared by a strong synoptic southwesterly flux. Unfortunately, we do not have ground or satellite observations to confirm this feature. As shown in Figure S11, Bhutan

is often covered by clouds over the north making the retrieve of Aerosol Optical Depth almost impossible most of days. For the last period, we performed a numerical simulation to highlight the role of forest fires over the West of Bhutan. In Figure 11, we compare the PM$_{2.5}$ concentration between a case without fires over the three domains (NOFI) and the reference simulation with wildfires (CTRL) over a 10 days period 17-31 March 2025. The difference $[CTRL - NOFI]$ can be then considered as the contribution of wildfires.

Over the widest domain, we observe that the main impact is located over the East part of the domain in Myanmar and East India (states of Assam, Meghalaya, Manipur, Mizoram and Nagaland). These regions are usually affected by wildfires in spring and particularly in March and April before the monsoon season (Unnikrishnan and Reddy, 2020). In this location, fires contributes to more than 40% of PM$_{2.5}$ in line with findings of Kumari et al. (2024). In the Hindu Kush Himalaya region, Nepal and Bhutan are the most affected and it is only the start of the fire season for these countries. Over the West Bhutan, the local

fires in Haa and Paro district contribute locally to up to 20% of PM$_{2.5}$ on average over the 15 days period. This average value hides large time variability with sometimes hourly contributions exceeding 70 $\mu$g m$^{-3}$ at close to wildfire emission sources. The contribution of fires issued from Myanmar an India can be observed in the South-East border of Bhutan with a relative contribution of 20% thanks to south-easterly winds. Removing fires also reduce other gas emissions leading to potential some small increases over the north of India due to non-linearities on chemistry, the slight increase of PM is mainly due to some

increase of sulfate formation without fires over some places in India and over the Indian ocean. The sulfur chemistry and sulfate formation is influenced by the oxidant capacity and the pH of cloud droplets, then a small change can positively or negatively modify all chemical processes slight increase of sulfate. Thus, it is not surprising to observe the strongest changes over the ocean where the relative humidity is high and where sulfate is also emitted by sea salts and produced from the Dimethyl sulfide.

Wild fires can contribute up 10-20 % during some hours at Katsho. The contribution is mainly driven by carbonaceous

species (black carbon and organic matter). Interestingly, all statistics are improved when fires are taken into account with a slight improvement of time correlation, 0.60 and 0.66 respectively for NOFI and CTRL simulations. Wildfires have also an influence on ozone, NO$_2$, sulfate and nitrate concentrations (Figure S12). On overall, without fire emissions the concentrations are impacted by up to -10 ppb in the east part of the wider domain THP025. This indirect impact is due to the NO$_2$ emission





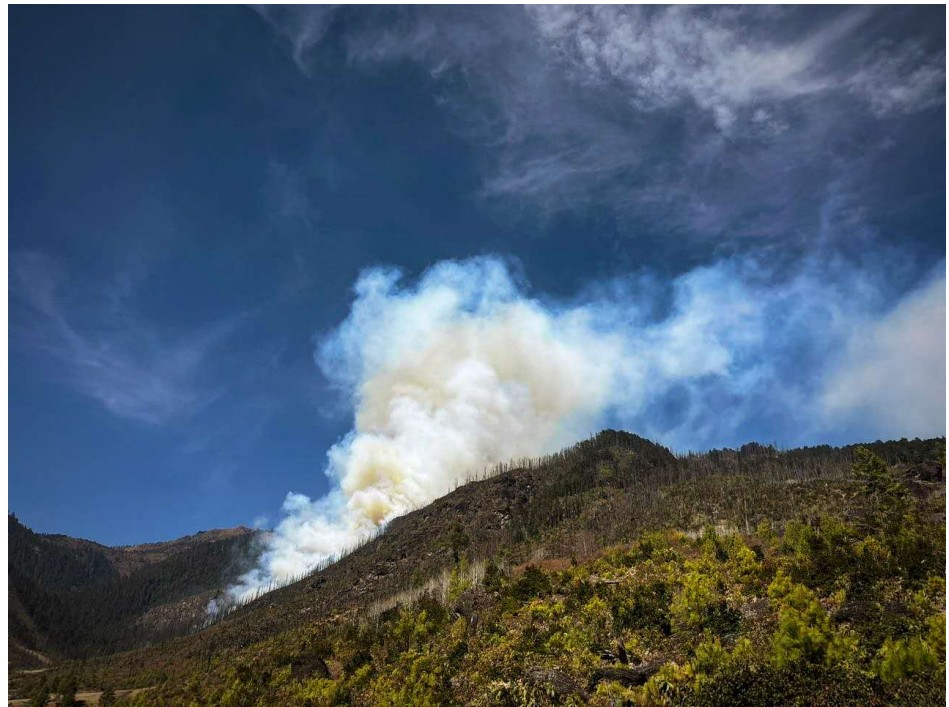

**Figure 10.** Wildfires observed nearby Haa on 21 March 2025 (*credit: Tenzin Wangchuk*)

reductions without wildfires in a NOx-limited region (low NOx concentrations). Far from wildfire sources, $NO_2$ concentrations
can even slightly decrease. The chemistry is actually changed over the whole domain and in some place the oxidant capacity
can be slightly enhanced when the fire emissions are not taken into account. Because of specific chemistry regimes (due to
titration by NOx), an increase of ozone is rarely observed over the North of India leading in this region where the presence of
megacities leads sometimes to VOC-limited regimes. In Bhutan, a decrease of 1 to 2 ppb is observed, this region has low $NO_2$
concentrations and is mainly NOx-limited, and a decrease of Volatile Organic Compounds and NOx emissions due to wildfires
will automatically reduce ozone concentrations.

## 8 Conclusions

For the first time, an quality simulation at 1km horizontal resolution over Bhutan has been performed with the regional chemistry transport model CHIMERE over a large domain covering several Himalayan valleys. The model was run over a period
after the winter season (begin of February to end of March 2025) to benefit from additional measurements in Haa valley with
two low cost sensors. The simulations highlight the issue of air quality in Bhutan valleys with an important contribution of
carbonaceous species to the level of $PM_{2.5}$ concentrations that largely exceed the WHO guidelines. From the analysis of the
simulation outputs, we can draw the additional following conclusions:





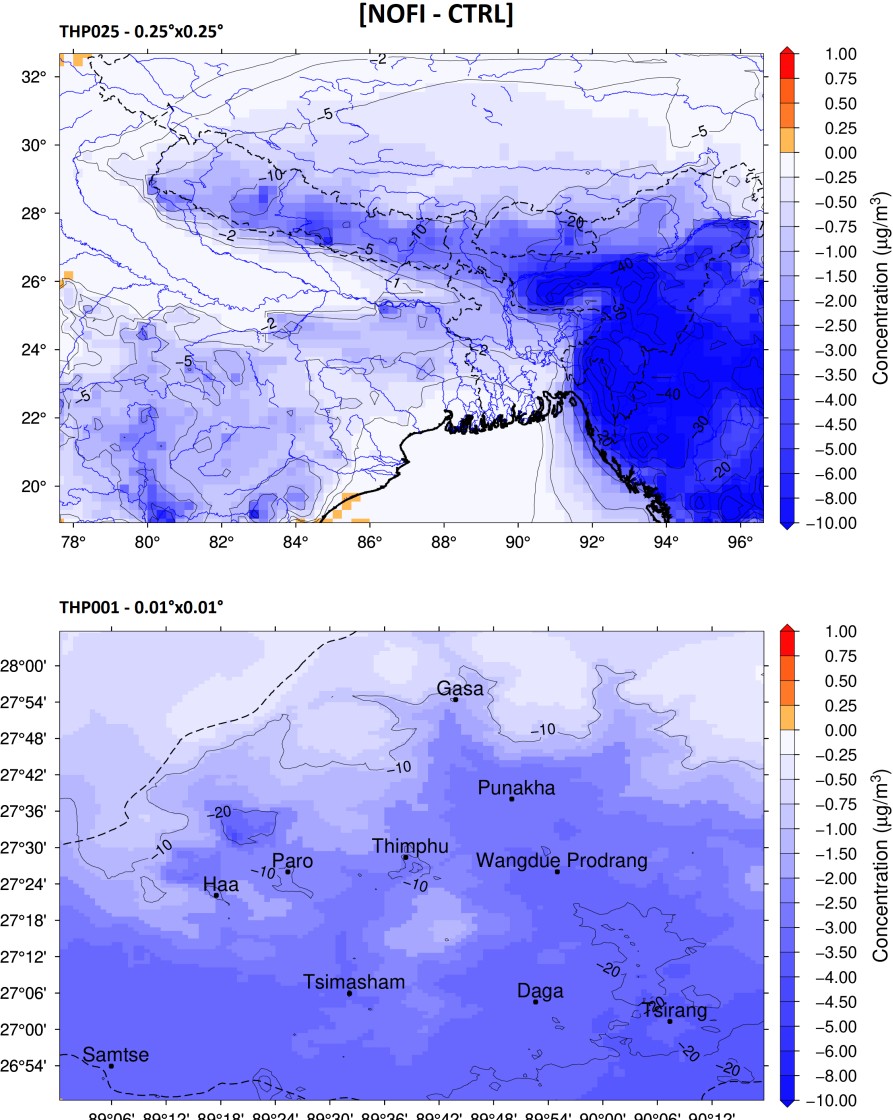

**Figure 11.** Impact of fires on PM$_{2.5}$ concentrations for the THP025 and THP001 domains (respectively top and bottom). Average concentration decrease without fire emissions over all domains (NOFI - CTRL) from the 17 to 31 March. The contour lines are the reduction in % (from the CTRL reference case concentration)

- – increasing the spatial resolution improves the model performances on meteorological variables and air pollutant concentrations in urban areas, but improvements of the spatialisation is still needed particularly for industrial sources,

– road dust resuspension must be considered in models as it is an important fraction of coarse particles but also fine particles, particularly in South-Asia,




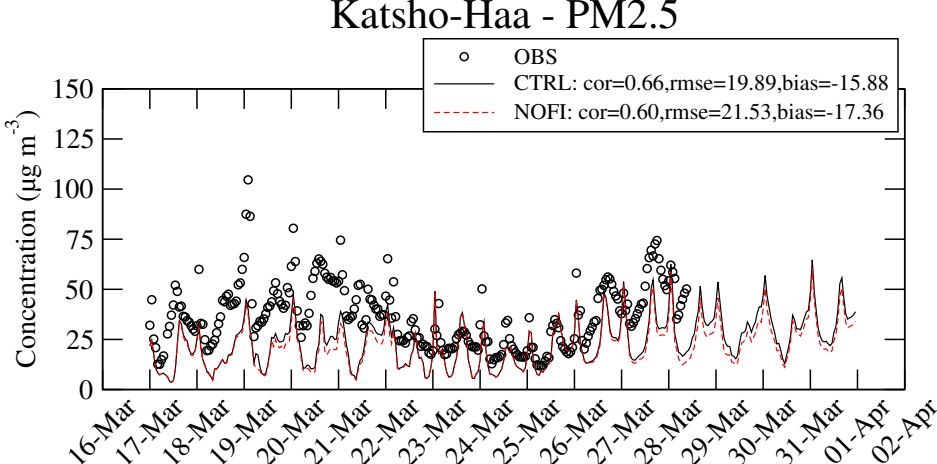

**Figure 12.** Timeseries in UTC time of PM$_{2.5}$ concentrations for the CTRL reference simulation and the simulation without fires (NOFI) against observations

- cumulative total deposition fluxes in March 2025 over the finest domain at $0.01^o$ of key species over the month of March 2025 are consistent through the three spatial model resolutions with similar simulated values,

- the simulated patterns of deposition fluxes confirm the potential role of dust, BC and probably BrC on glaciers melting and the capacity of the model to simulate the impact of air pollution,

- our simulations confirm that Himalayan forests are expected at risk from excess nitrogen, the levels of deposition fluxes calculated in March are comparable to high values computed or observed in many other locations worldwide,

- wildfires can have a strong impact in the valleys of Bhutan, their contribution to PM$_{2.5}$ concentrations can exceed averaged values of 20% during outbreak of fires and Bhutan can also be affected from fires occuring in North-East India and Myanmar.

At this point, a model setup to simulate air quality is available. Additional studies and applications can be carried out to monitor air pollution and create the most effective measures for enhancing air quality in the Himalayan valleys. This study is an opportunity to highlight the need to have a more comprehensive modelling framework to study the impact on glaciers accounting for dust and the carbon from the Black and Brown Carbon, this latter been driven by biomass burning sources that are important in the Himalayan valleys. For PM$_{2.5}$, it is also important to create a network of low-cost sensors in the Hindu Kush Himalaya to monitor both indoor and outdoor air pollution, complementing a set of reference stations. Increasing the resolution will be helpful to evaluate the effect of air pollution on glaciers, ecosystems, and human health including social aspects. Then, using cutting-edge machine learning techniques, models will be used in combination with measurements to get an exhaustive picture of air pollution in the region and participate to the necessary dialogues between all the countries in the Hindu Kush Himalaya..





*Code availability.* The CHIMERE v2023r1 model is available on Zenodo at https://doi.org/10.5281/zenodo.10907951

*Data availability.* All data generated for this study can be sent on request

*Code and data availability.* The data required for the code are freely available at https://www.lmd.polytechnique.fr/chimere/2023_getcode.php

*Video supplement.* A 3D video is provided to show the evolution of the smoke plumes due to wildfires reaching the Jomolhari peak

*Author contributions.* BB conceptualized and piloted the study, ran the model, analyzed the results and wrote the draft paper; NT has developed the proxy for the emissions; DPB, RS and TW deployed the low cost sensors in Haa; AS, AC, LM and GS supported the set-up of the modeling chain, MC supported on the global emission inventory; KG piloted the overall project related to the sensor deployments. All authors participated to the review of the paper and the analysis of data.

*Competing interests.* The contact author has declared that neither of the authors has any competing interests.

*Acknowledgements.* This work was granted access to the HPC resources of TGCC operated by GENCI. The authors thank the National Center for Hydrology and Meteorology (NCHM) of the Royal Government of Bhutan for giving access to official meteorological data. ICIMOD staff is being supported by the United Kingdom's Foreign, Commonwealth and Development Office (FCDO) under their Climate Action for Resilient Asia (CARA) initiative.



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
