# Peer review of "High resolution Air Quality simulation in the Himalayan valleys, a case study in Bhutan"

_EGUsphere, 2025_

## Author Comment (AC1)

**Replies to Reviewer 1**

We would like to thank the reviewer for her/his positive and constructive feedback, which helped to improve the quality of the paper. Below we address specific issues mentioned by the reviewers point by point. The manuscript has been updated accordingly. We have taken the liberty of making minor stylistic edits elsewhere in the manuscript.

**General comments**

The paper High resolution Air Quality simulation in the Himalayan valleys, a case study in Bhutan presents an application of the CHIMERE chemistry transport model over the west Bhutan. Though the modelling application is quite standard the area of study is particularly challenging due to the scarcity of data and available literature.

The authors provided a thorough description of the modelling design and implementation as well as of the evaluation of the obtained results, though limited by the availability of observed meteorological and air quality data.

A second interesting aspect of this paper is the evaluation and discussion of different issues related to the effect of air pollution in remote areas, such as deposition over glaciers and role of wildfires.

Therefore, the paper fits the scope of ACP. The paper is also well written, with concise and clear statements, and it does not require any substantial review of syntax and language.

The paper could be published considering just a general review of the section on the evaluation of the model performance that is sometimes unclear and partially confusing.

To this aim, additional details are available in the following section.

**Specific comments and Technical corrections**

P4 - R97-98 - Sentence is not clear

Corrected we have clarified there was a small mistakes.

P8 – R183 – Are observed data discussed in this subsection presented in Figure S2?

P10 – R202 - Are observed data discussed in this subsection presented in any figure?

For both comments, yes we refer in the paragraph to Figure S2-S4

P10 – R219 – Does Figure S5 refer to all available data?

Indeed, in Figure S5 we have plotted the diurnal cycles of all variable whenever it was possible.

P11 - R229 - A relation exists "between the observed PM coarse fraction" and what?

Perfectly right, we have forgotten a piece of text "... with the relative humidity..."

P11 – r230-235 – This section is not very clear. This paragraph should be focused on model performance evaluation, but here the discussion seems on observed data, which are also compared to literature data

We understand this point. Actually, in this paragraph we talk only about measurements but to explain a potential underestimation of PM concentrations. However we have moved this section in the "Available Observation" section.

P11 - R238-241 – Discrepancies in Haa stations for PM2.5 during March seem more related to a difficulty of the model in capturing the two episodes (Meteorology? Emissions?) than to spatial resolution

Yes we agree, we have added: "For Haa, the model has poor performances for PM2.5 concentrations which is probably due to more general difficulties to simulate the meteorology in this altitude valley and/or the emissions from natural and anthropogenic origins."

P11 - R250 - "overestimates"

True, we have corrected.

P11 - R253-255 - Is BC time series shown in Figure 5?

Yes, we have clarified.

P12 – Figure 4 – Is this Figure mentioned in the text?

We agree, the figure was not mentioned. Now, it is mentioned and note that the whole paragraph has been moved.

P12 - R258-262 - Are the industrial sources considered as point sources or ground level emissions? Could Also this aspect influence the performance?

Yes we have added this statement: "As in many global and regional inventories, in our case, the source points are actually merged into surface emission with raw estimates of injection heights based on vertical emission profiles (Bieser, 2011)". This has no impact on the model performances since the eulerian model is, by definition, a model where all sources are mixed into a volume.

P12 – R262-266 – Why do the analysis of meteorological performance is not placed before air quality?

We have used the analysis of meteorological data performance to explain the potential issues on air quality that is why it comes after.

P14 – R294 – contour lines in Figure 8 are visible only over white areas

Yes the contour line are sometimes difficult to observe but with a good resolution quality it is perfectly readable by increasing the size of the figure. We tested several options and this one was the most adequate.

P15 - R318 - "1 or 2 mg/m2"?

True, it is in mg/m2, we have corrected

P21 - R380 - How were wildfire emissions estimated and modulated?

We have clarified in the model setup section: "Fire emissions fluxes are calculated from daily CAMS Global Fire Assimilation System (GFAS, 2022). The data used are the analysis and have an horizontal resolution of 0.1 × 0.1 degrees (Kaiser et al., 2012). These data are reformatted to be consistent with the required CHIMERE input data. This procedure includes: (1) spatially, the data are projected on the CHIMERE horizontal model grid, (2) temporally, these daily data are interpolated to provide hourly fire emission fluxes, (3) chemically, the chemical species are disaggregated and reaggregated to be consistent with the model chemical mechanism. The injection height is calculated using the scheme of Sofiev et al. (2012) requiring the Fire Radiative Power (in Mega Watts). Vertically, the shape of the emission profile is calculated using a profile defined and described in Menut et al. (2018). The accumulation of burnt areas is calculated and allows to calculate the impact of fires on the surface then mineral dust emissions, biogenic emissions and dry deposition as explained in Menut et al. (2022, 2023)"

We refer to the CHIMERE documentation.

**High resolution Air Quality simulation in the Himalayan valleys, a case study in Bhutan**

Bertrand Bessagnet1, Narayan Thapa2, Dikra Prasad Bajgai2, Ravi Sahu2, Arshini Saikia2, Arineh Cholakian1, Laurent Menut1, Guillaume Siour3, Tenzin Wangchuk4, Monica Crippa5, and Kamala Gurung2

**Correspondence:** Bertrand Bessagnet (bertrand.bessagnet@lmd.ipsl.fr)

Abstract. Our study focuses on Bhutan, a highly mountainous country where governmental authorities are increasingly monitoring air pollution government authorities are strengthening air pollution monitoring efforts. To support further analysis and the monitoring strategy, we present the first high-resolution air quality simulations with the chemistry transport model WRF-CHIMERE over the western region of Bhutan at a spatial resolution of roughly 1 km. Increasing the horizontal resolution of the model improve the performances, decreases potential errors due to too important spatial average improves its performance and reduces potential errors caused by excessive spatial averaging of meteorological and emissions data having an emission data with high spatial variability. However, the air pollutant emissions must be improved at a fine scale with better proxy, particularly for industries where improvement improvements are still required. For the first time, we propose high resolution maps of air pollution (concentrations and deposition fields). Our simulations confirm that Bhutan valleys also suffer from air pollution mainly due to PM2.5 (concentrations exceeding 20  $\mu$ g m-3) dominated by carbonaceous species, largely above the World Health Organization guidelines. Wildfires and anthropogenic activities release large amount of carbonaceous species and can also impact the glaciers by atmospheric fallout. Wildfires can locally contribute to 20% of the total PM2.5 concentrations over a 15 days period, and theoretically, black carbon can be transported up to the highest peaks. Ecosystems are at risks with deposition fluxes of sulfur and nitrogen species comparable with other locations at risk in the world.

**15 1 Introduction**

The Hindu Kush Himalaya (HKH) spans over a region particularly affected by air pollution including India, Pakistan, Bangladesh and Nepal which are currently the most impacted by air pollution (HEI, 2025; Mehra et al., 2019). Particularly outside the monsoon season, the combination of favorable meteorological conditions and large emission sources in the Indo-Gangetic Plain is the main reason of impressive outbreak of pollution events. In the region, air pollution not only affects health (HEI, 2025). Air pollution also has an effect on ecosystems (loss of biodiversity) through the deposition of inorganic species like ammonia

<sup>1Laboratoire de Météorologie Dynamique (LMD)/IPSL, Ecole Polytechnique, Institut Polytechnique de Paris, ENS, Université PSL, Sorbonne Université, CNRS, Route de Saclay, 91128 Palaiseau, France

<sup>2International Centre for Integrated Mountain Development (ICIMOD), Kathmandu, Nepal

<sup>3Univ Paris Est Creteil and Université Paris Cité, CNRS, LISA, F-94010 Créteil, France

<sup>4Jigme Singye Wangchuck School of Law, Paro, Bhutan

<sup>5European Commission, Joint Research Centre, Ispra, Italy

[revised manuscript text omitted]

---

## Author Comment (AC2)

**Replies to Reviewer 2**

We would like to thank the reviewer for her/his positive and constructive feedback, which helped to improve the quality of the paper. Below we address specific issues mentioned by the reviewers point by point. The manuscript has been updated accordingly. We have taken the liberty of making minor stylistic edits elsewhere in the manuscript.

**General comments**

This manuscript presents the first high-resolution (1 km) regional air-quality simulation over western Bhutan using the WRF-CHIMERE modelling system. The authors develop a new fine-scale emission inventory—downscaled from EDGAR—and simulate PM2.5, O3, and other pollutant fields for February-March 2025. Model outputs are evaluated against sparse observations (the Thimphu reference station and two low-cost sensors in the Haa valley), and spatial patterns of air pollution and deposition are examined. Key findings include persistently elevated PM2.5 levels that exceed WHO guidelines, with carbonaceous aerosols dominating the load; substantial contributions from biomass burning (wildfires); and non-negligible long-range transport of BC and dust affecting high-altitude Himalayan glaciers. The study further quantifies nitrogen, sulfur, carbon, and dust deposition, showing that fluxes—especially for nitrogen species—are comparable to values observed in high-risk ecosystems globally. While increased spatial resolution generally improves model performance (notably for temperature and near-surface gradients), several biases remain. Overall, this work provides valuable high-resolution pollution maps and new insights into the drivers of air quality in Bhutan, with implications for regional environmental management. The novelty lies less in the modelling methodology itself than in the application to a region that has been seldom studied and is characterized by complex orography and diverse emission sources.

**Major Comments**

Conducting air-quality simulations at 1 km resolution over Bhutan's highly complex terrain is both novel and useful. As the authors note, no comparable fine-scale modelling has been performed in this region. The study addresses relevant questions on local versus transported pollution, the role of wildfires, and deposition processes. However, the analysis is limited to a  $\sim$ 1.5-month late-winter to early-spring period. The manuscript would benefit from a brief discussion of how representative these months are of typical annual conditions, or from an explicit acknowledgement of the limitations of this single-season case study.

We agree, we have highlighted now this limitation in the conclusions: "The model has been run over a limited temporal window which is the main limitation of this study."

The custom high-resolution inventory is a strength of the study. The proxy-based downscaling of EDGAR emissions is clearly described, and the dominance of residential combustion aligns with known local practices (e.g., wood-burning stoves). Nevertheless, this approach inevitably overlooks some local sources and spatial heterogeneity. Industrial emissions, in particular, remain highly uncertain. The manuscript would be strengthened by an explicit discussion or estimation of the uncertainties associated with these proxies and by a qualitative assessment of the sensitivity of the results to downscaling assumptions.

We highlight now the issue of industries and particularly one of the most spread in this region: "One peculiarity in the region is the Brick Kiln industry which is a major issue for the environment and human health. The brick kiln sector across South Asia, including the Himalayan region, operates largely in the informal economy, making it hard to regulate or relocate. Kilns are often scattered, unregistered,

and invisible to policymakers, which complicates efforts to enforce environmental or labor standards. The stack are so complicated to localize that it is very challenging for the emission community to correctly account for this source (Tahir et al., 2021, Das et al., 2025). This specific sector is not isolated in EDGAR, then we reallocate spatially the emissions of this sector with a single industrial proxy from OSM."

In the conclusions we have now added: "At last, the creation of a high resolution emission inventory with a bottom-up approach should be a priority of the regional institutions responsible for air quality management"

The nested WRF–CHIMERE setup (0.25°, 0.05°, 0.01°) is well described, and the use of 46 vertical levels with nudging in the outer domain is appropriate. The choice of the simulation period is justified by available observations. For completeness and reproducibility, the authors should consider adding a succinct table or flow chart summarizing key model configuration aspects—e.g., boundary conditions, chemical mechanisms, emission species. In addition, the manuscript should further clarify how the chosen vertical resolution improves representation of valley inversion layers and cold-pool dynamics.

For completeness and reproductibility the WRF and CHIMERE namelists are provided in Table S1 and S2 in the supplementary material. We agree with the reviewer, we think that it should be the case for any paper related to modelling studies. The WRF namelist is inspired by the one used in Bessagnet et al., 2020. We have added a statement referring to this paper "The various WRF parameters are similar of those used in Bessagnet et al., 2020 where a simulation over a complex topography was performed with an evaluation of the vertical patterns."

The analysis of fire impacts is compelling. The comparison between CTRL and NOFI configurations shows that local fires in March increased  $PM_{2.5}$  by  $\sim 20$  % in affected districts, with hourly peaks exceeding 70 %, while fires in Myanmar and India contribute  $\sim 20$  % near Bhutan's southeastern border. These results are policy-relevant. However, the treatment of fire emissions should be described more precisely: Were CAMS GFAS emissions used exclusively? Were local hotspots incorporated? It would also be useful to comment on the degree to which the model reproduces the observed spatial and temporal patterns of the wildfire events.

Based on the previous comment of reviewer 1 we have extended the explanation of the treatment of fires in CHIMERE that are fully explained in the documentation. Only wildfire emissions from CAMS were used but they accurately spotted the Bhutan fires for our episodes. In Figure S10 we show some retrieval from satellites showing the extent of fires from the Visible Infrared Imaging Radiometer Suite.

Mapping deposition fluxes is highly relevant for evaluating glacier darkening and ecosystem vulnerability. The manuscript convincingly shows that nitrogen and organic-carbon deposition reaches levels comparable to sensitive regions worldwide. To contextualize these results, the authors could briefly compare their values with known ecosystem critical loads or with findings from similar Himalayan studies.

A critical load is a quantitative estimate of pollutant exposure (e.g., nitrogen, sulfur, heavy metals, etc..) below which significant harmful effects on sensitive ecosystem components do not occur. A critical load is computed by models using (i) soil chemistry (buffering capacity, base saturation), hydrology (water flow, retention) and vegetation sensitivity. We did not compute critical loads in our study, it could be the next step. However, in Ellis et al. (2022) they provide an estimate for N, and to our knowledge it is the only one for a pollutant in this region.

The reviewer points out some approximations in the discussion and then helps us to reformulate as :"The average value of the total deposition flux of nitrogen for the month of March 2025 (around 300 gN  $ha^{-1}$  on average as reported in Table 4) is of the same order of magnitude with a critical value estimated by Ellis (2022) at 4.24 kgN  $ha^{-1}$  yr-1 for the Himalayan region. According to the deposition fluxes displayed in Figure 8 (sometimes exceeding 100 mgN  $m^{-2}$  i.e. 1 kgN  $ha^{-1}$  for one month) this annual threshold is likely to be exceeded in the South of Bhutan. However, this needs to be assessed

with a proper calculation of the critical loads with an annual simulation. Indeed, A critical load requires the use of models using (i) soil chemistry (buffering capacity, base saturation), hydrology (water flow, retention) and vegetation sensitivity. For Europe, deposition values of more than  $1 \text{ kgN ha}^{-1} \text{ yr}^{-1}$  were simulated by (Vivanco, 2017)".

The study clearly highlights that  $PM_{2.5}$  levels in valleys and urban centers routinely exceed health guidelines, informing mitigation priorities such as cleaner residential fuels and wildfire management. As a second step, the authors are encouraged to implement ensemble modelling approaches—through multi-physics, multi-parameter, or multi-model perturbations—to quantify structural and parametric uncertainties inherent in high-resolution atmospheric simulations. Ensemble frameworks provide probabilistic ranges instead of single deterministic outputs, thereby reinforcing the robustness of the findings and greatly enhancing their utility for evidence-based policy and decision-making.

Thank you for this suggestion. We totally agree that, for the region, we need a multi-model approach to get more robust conclusions on the best air pollutant emission reduction strategies to adopt in the region. The first author spent a while in the HKH at ICIMOD and promoted through this institution the development of a regional cooperation to tackle air pollution: https://blog.icimod.org/air/clean-air-dialogue-thimpu/. This modelling work will support this initiative which was cited in the introduction.

**Minor Comments**

Wording issue: "an quality simulation" should be corrected to "an air-quality simulation."

We have corrected.

In the emission inventory section, clarify that "primary OM" refers to organic matter, as this may not be familiar to all readers, particularly those less accustomed to EDGAR conventions.

We have clarified.

**High resolution Air Quality simulation in the Himalayan valleys, a case study in Bhutan**

Bertrand Bessagnet1, Narayan Thapa2, Dikra Prasad Bajgai2, Ravi Sahu2, Arshini Saikia2, Arineh Cholakian1, Laurent Menut1, Guillaume Siour3, Tenzin Wangchuk4, Monica Crippa5, and Kamala Gurung2

**Correspondence:** Bertrand Bessagnet (bertrand.bessagnet@lmd.ipsl.fr)

Abstract. Our study focuses on Bhutan, a highly mountainous country where governmental authorities are increasingly monitoring air pollution government authorities are strengthening air pollution monitoring efforts. To support further analysis and the monitoring strategy, we present the first high-resolution air quality simulations with the chemistry transport model WRF-CHIMERE over the western region of Bhutan at a spatial resolution of roughly 1 km. Increasing the horizontal resolution of the model improve the performances, decreases potential errors due to too important spatial average improves its performance and reduces potential errors caused by excessive spatial averaging of meteorological and emissions data having an emission data with high spatial variability. However, the air pollutant emissions must be improved at a fine scale with better proxy, particularly for industries where improvement improvements are still required. For the first time, we propose high resolution maps of air pollution (concentrations and deposition fields). Our simulations confirm that Bhutan valleys also suffer from air pollution mainly due to PM2.5 (concentrations exceeding 20  $\mu$ g m-3) dominated by carbonaceous species, largely above the World Health Organization guidelines. Wildfires and anthropogenic activities release large amount of carbonaceous species and can also impact the glaciers by atmospheric fallout. Wildfires can locally contribute to 20% of the total PM2.5 concentrations over a 15 days period, and theoretically, black carbon can be transported up to the highest peaks. Ecosystems are at risks with deposition fluxes of sulfur and nitrogen species comparable with other locations at risk in the world.

**15 1 Introduction**

The Hindu Kush Himalaya (HKH) spans over a region particularly affected by air pollution including India, Pakistan, Bangladesh and Nepal which are currently the most impacted by air pollution (HEI, 2025; Mehra et al., 2019). Particularly outside the monsoon season, the combination of favorable meteorological conditions and large emission sources in the Indo-Gangetic Plain is the main reason of impressive outbreak of pollution events. In the region, air pollution not only affects health (HEI, 2025). Air pollution also has an effect on ecosystems (loss of biodiversity) through the deposition of inorganic species like ammonia

<sup>1Laboratoire de Météorologie Dynamique (LMD)/IPSL, Ecole Polytechnique, Institut Polytechnique de Paris, ENS, Université PSL, Sorbonne Université, CNRS, Route de Saclay, 91128 Palaiseau, France

<sup>2International Centre for Integrated Mountain Development (ICIMOD), Kathmandu, Nepal

<sup>3Univ Paris Est Creteil and Université Paris Cité, CNRS, LISA, F-94010 Créteil, France

<sup>4Jigme Singye Wangchuck School of Law, Paro, Bhutan

<sup>5European Commission, Joint Research Centre, Ispra, Italy

[revised manuscript text omitted]

---

## Author Response (AR2)

**Bertrand Bessagnet**

Directeur of Research - CNRS

Laboratoire de Météorologie Dynamique (LMD)

Ecole Polytechnique – Palaiseau

bertrand.bessagnet@lmd.ipsl.fr

*Objet : Second review of article egusphere-2025-3641*

Dear Editor,

Please find enclosed the review of our paper entitled "**High resolution Air Quality simulation in the Himalayan valleys, a case study in Bhutan**" to ACP.

As you have suggested, we have changed:

*"At last, the creation of a high resolution emission inventory with a bottom-up approach should be a priority of the regional institutions responsible for air quality management"*

by

*"As a last point, we suggest that the creation of a high resolution emission inventory with a bottom-up approach should be a priority of the regional institutions responsible for air quality management."*

On behalf of all co-authors, I thank you again to have edited our paper.

Yours sincerely,

Bertrand Bessagnet on behalf of all co-authors

On 6 December 2025, Palaiseau, France